# The Importance of Endoplasmic Reticulum Stress as a Novel Antidepressant Drug Target and Its Potential Impact on CNS Disorders

**DOI:** 10.3390/pharmaceutics14040846

**Published:** 2022-04-12

**Authors:** Marta Jóźwiak-Bębenista, Paulina Sokołowska, Małgorzata Siatkowska, Cecilia Analia Panek, Piotr Komorowski, Edward Kowalczyk, Anna Wiktorowska-Owczarek

**Affiliations:** 1Department of Pharmacology and Toxicology, Medical University of Lodz, Zeligowskiego 7/9, 90-752 Lodz, Poland; paulina.sokolowska@umed.lodz.pl (P.S.); edward.kowalczyk@umed.lodz.pl (E.K.); anna.wiktorowska-owczarek@umed.lodz.pl (A.W.-O.); 2Laboratory of Molecular and Nanostructural Biophysics, Bionanopark, Dubois St. 114/116, 93-465 Lodz, Poland; m.siatkowska@bionanopark.pl (M.S.); p.komorowski@bionanopark.pl (P.K.); 3Laboratory of Personalized Medicine and Biotechnological Laboratory, Bionanopark, Dubois St. 114/116, 93-465 Lodz, Poland; c.panek@bionanopark.pl; 4Division of Biophysics, Institute of Materials Science, Lodz University of Technology, Stefanowskiego 1/15, 90-924 Lodz, Poland

**Keywords:** antidepressant drugs, CHOP, depression, ER stress, ketamine, OASIS, UPR pathway

## Abstract

Many central nervous system (CNS) diseases, including major depressive disorder (MDD), are underpinned by the unfolded protein response (UPR) activated under endoplasmic reticulum (ER) stress. New, more efficient, therapeutic options for MDD are needed to avoid adverse effects and drug resistance. Therefore, the aim of the work was to determine whether UPR signalling pathway activation in astrocytes may serve as a novel target for antidepressant drugs. Among the tested antidepressants (escitalopram, amitriptyline, S-ketamine and R-ketamine), only S-ketamine, and to a lesser extent R-ketamine, induced the expression of most ER stress-responsive genes in astrocytes. Furthermore, cell viability and apoptosis measuring assays showed that (R-)S-ketamine did not affect cell survival under ER stress. Under normal conditions, S-ketamine played the key role in increasing the release of brain-derived neurotrophic factor (BDNF), indicating that the drug has a complex mechanism of action in astrocytes, which may contribute to its therapeutic effects. Our findings are the first to shed light on the relationship between old astrocyte specifically induced substance (OASIS) stabilized by ER stress and (R-)S-ketamine; however, the possible involvement of OASIS in the mechanism of therapeutic ketamine action requires further study.

## 1. Introduction

The endoplasmic reticulum (ER), the largest organelle in eukaryotic cells, is a dynamic structure that regulates multiple biological processes, particularly the synthesis, folding, maturation and transport of proteins. Moreover, the ER is responsible for calcium storage and the biosynthesis of lipids and sterols [1,2]. ER dysregulation can be triggered by internal and external factors, resulting in the accumulation of unfolded and misfolded proteins in the ER lumen, known as ER stress. To counteract this harmful effect and prevent apoptosis, the unfolded protein response (UPR) pathway is activated.

The UPR pathway includes the activation of three ER transmembrane effector proteins: inositol-requiring enzyme 1 (IRE1) [3,4], protein kinase R-like ER kinase (PERK) [5] and the activating transcription factor 6 (ATF-6) [6]. Under physiological conditions, these three proteins are stored in inactive forms by associating with reticular chaperones: 78-kDa glucose-regulated protein (GRP78) and 94-kDa glucose-regulated protein (GRP94) [7]. When ER stress is induced, GRP78 and GRP94 disassociate from PERK, IRE1 and ATF-6, thereby, activating their intracellular anti-apoptotic and/or pro-apoptotic functions [8,9,10,11,12] (Figure 1).

Multiple studies have demonstrated that dysregulation or hyperactivation of the UPR pathway is involved in the pathophysiology of various conditions, such as cardiovascular disease, cancer and metabolic disease [13,14]. It has also been shown that ER stress is associated with neurodegenerative diseases (Alzheimer’s disease and Parkinson’s disease) [14] and other central nervous system (CNS) disorders, such as schizophrenia [15] or mood disorders, such as bipolar disorder [16] and major depressive disorder (MDD) [17,18].

MDD, often referred to as depression, is a common illness believed to affect an estimated 3.8% of the global population. The World Health Organization (WHO) indicates that depression is the fourth leading contributor to the global burden of disease, with the incidence of depression and anxiety growing due to lifestyle or the COVID 19 pandemic. This fact, together with the frequent loss of effectiveness of available antidepressant drugs during long-term treatment of depression, unbearable side effects [19] and patient unresponsiveness [20], indicates a need for new antidepressant drugs with novel mechanisms of action.

The mechanisms of action of currently used drugs in the management of MDD are based mainly on monoamine theory suggesting that depression arises in response to deficiencies in inter alia serotonin and /or norepinephrine neurotransmitters in the CNS [21,22]. However, it is now believed that their antidepressant effect results not only from elevated monoamine neurotransmission but also from modulatory action on neuroplasticity by stimulating the signalling pathways responsible for neuronal adaptations [23].

Indeed, there are numerous findings regarding the pathophysiology of depression, which may become new drug targets, such as abnormalities in glutamatergic neurotransmission, neuroendocrine theory, inflammation or neurotrophic conception based on the reduced expression of brain-derived neurotrophic factor (BDNF) and/or the failure of its receptor—tyrosine receptor kinase B (TrKB) [24,25,26,27]. Although neuronal atrophy has been highlighted to underlie the structural and cognitive manifestations of MDD, the loss of glial cells in the cognitive dysfunction observed in mood disorders increasingly appears to play a role.

Studies in postmortem brain samples from depressed patients demonstrated lowered density and number of glial cells in cortical regions, most relevantly in the prefrontal and cingulate cortex areas. [21,28,29]. It is important to note that astrocytes are a major class of glial cells that play a critical role in maintaining proper neuronal activity, preventing oxidative stress and enabling basic brain functions [30]. Astrocytes participate in the reuptake of norepinephrine and serotonin (the cells express transporters for both norepinephrine–NET and serotonin–SERT), metabolism and recycling of glutamate and provide support to neurons by secreting various trophic factors (e.g., BDNF and VEGF).

There is abundant evidence suggesting a direct role of astrocytes in the pathophysiology of depression and in antidepressant effects related to changes in neuronal plasticity [21]. This raises a possibility that astrocytes can be considered a target cell type for antidepressants mainly by releasing plasticity-enhancing growth factors; specifically, the role of astrocytic BDNF has been in focus. However, mechanisms explaining the astrocytic involvement in the action of antidepressants need to be uncovered.

The UPR pathway is of particular interest in depression. Many animal models of depression and human studies show an elevated brain ER stress response. In studies on rats, Zhang et al. found that the mRNA and protein levels of GRP78, XBP1 and CHOP were significantly increased after 21 continuous days of restraint stress (RS)—a common animal model for the production of chronic stress. This suggested that stress related to the ER is associated with the damage to the hippocampus and cognitive impairment [31].

Similar observations were obtained by Liu et al. in that the expression of GRP78 and XBP1 were essentially enhanced in the hippocampus of mice with social defeat stress [32]. A link between ER stress and depression was also achieved in human studies. Bown et al. [17] showed higher levels of GRP78, GRP94 and calreticulin in the temporal cortex of patients with MDD who died by suicide, compared with patients with non-suicide deaths and subjects who died from other reasons [17].

Another research group reported elevated and persistent systemic expression of ER stress-related genes in peripheral tissues of patients with MDD. An analysis of leukocyte-derived RNA samples found significantly higher levels of GRP78, EDEM1, CHOP and XBP1 in patients with MDD compared to control patients [18]. These results suggest that dysregulation of the endoplasmic reticulum lumen has an impact on the pathophysiology of major depressive disorders and may be a target for the treatment of this disease. Recently, several preclinical and clinical studies have indicated a strong relationship between depression and changes in the ER [33,34]. However, the effects of existing antidepressant drugs on the UPR pathway remain unclear [35,36].

Guidelines for treating depression suggest the use of drugs acting on monoaminergic neurotransmission, such as selective serotonin reuptake inhibitors (SSRIs) and serotonin and norepinephrine reuptake inhibitors (SNRIs) among other drugs, including agomelatine, bupropion, mirtazapine and vortioxetine, as first-line drugs. In the case of ineffectiveness, antidepressants, such as tricyclic antidepressants (TCAs) or monoaminoxidase inhibitors (MAOI), are prescribed as second-line drugs, especially in patients with treatment-resistant depression (TRD). The new option for TRD patients is S-ketamine, which influences the glutamatergic system. Although antidepressant drugs are well-known for their therapeutic efficacy in MDD, their exact mechanism of action, especially that of ketamine, is still being discussed.

Therefore, the aim of this study was to examine whether, and to what extent, commonly used antidepressant drugs and S-ketamine (approved by the FDA for the treatment of refractory depression since 2019) affect the expressions of genes related to the UPR signalling pathways in human astrocyte cell lines. Astrocytes were shown to play an essential role in neuronal plasticity disrupted in depression; therefore, ER stress in astrocytes may affect the functioning of neurons. The study uses antidepressants with different mechanisms of action, such as amitriptyline (a TCA), escitalopram (SSRI) and S-ketamine, whose the mechanism of action in depression is not fully understood. For comparison, the study also included R-ketamine—a less active enantiomer of ketamine currently tested in clinical trials.

## 2. Materials and Methods

### 2.1. Reagents

Tunicamycin, MTT, Hoechst 33342 solution, Tween 20, Trypsin-EDTA solution were purchased from Sigma-Aldrich (Saint Louis, MO, USA) and amitriptyline, escitalopram, R-ketamine and S-ketamine from Tocris (Ellisville, MO, USA). Dulbecco’s Phosphate Buffered Saline (D-PBS) was obtained from Biowest (Nuaillé, France). Astrocyte medium, Astrocyte Growth Supplement, Fetal Bovine Serum, penicillin/streptomycin solution and poly-L-Lysine were from ScienCell Research Laboratories (Carlsbad, CA, USA). Annexin V-FITC Fluorescence Microscopy Kit was from BD Pharmingen (Franklin Lakes, NJ, USA). NucleoSpin RNA kit was from Macherey-Nagel GmbH & Co. KG (Dueren, Germany), and High Sensitivity RNA ScreenTape Kit was from Agilent Technologies (Santa Clara, CA, USA).

Custom PrimePCR™ Real-Time PCR Plates, iScript™ cDNA Synthesis Kit, 2xSsoAdvanced Universal SYBR Green Supermix, Prime PCR RT Control, Prime PCR Control, Mini-Protean TGX Stain-Free Gels, 10× Tris/Glycine/SDS Buffer, Trans-Blot Turbo RTA Transfer Kit, Nitrocellulose, Trans-Blot Turbo 5× Transfer buffer, Clarity Western ECL Substrate Assay, 4× Laemmli Sample Buffer, 2-Mercaptoethanol, Precision Plus Protein All Blue Standards and Precision Plus Protein Unstained Standards were purchased from Bio-Rad (Berkeley, CA, USA). Reagents for lysis buffer (urea, thiourea, Tris, 3-[(3-Cholamidopropyl)dimethylammonio]-1-propanesulfonate hydrate (CHAPS), IPG Buffer pH 4–7 and dithiotreitol (DTT)) and 2D Quant Kit, 2D Clean Up Kit were purchased from GE Healthcare (Little Chalfont, United Kingdom). Anti-DDIT3 and anti-mouse IgG (HRP) were from Abcam (Cambridge, MA, USA).

### 2.2. Cell Culture 

The studies were performed on a commercially-available (Cat no. 1800, ScienCell Research Laboratories; San Diego, CA, USA) astrocyte cell line isolated from human brain tissue (cerebral cortex). Cell culture was maintained at 37 °C and in humidified atmosphere of 5% CO_2_. Cells were grown in Astrocyte Medium supplemented with 2% fetal bovine serum, 10% astrocyte growth supplement, 1% penicillin/streptomycin solution. The astrocyte culture was maintained according to the protocol recommended by ScienCell Research Laboratories.

### 2.3. MTT Test

Cell viability was evaluated using the colorimetric MTT assay. The cells were seeded a onto 96-well plate to a final density of 7 × 10^3^ cells/well. After 24 h of culture the cells were exposed to tunicamycin or tunicamycin with an antidepressant drug for the next 24 h. After the incubation time with drugs, MTT solution was added to the cell culture for another four hours. Mitochondrial succinate–tetrazolium reductase system converts yellow tetrazolium MTT (3-(4,5-dimethylthiazol-2-yl)-2,5-diphenyltetrazolium bromide) into purple formazan. The absorbance was measured at 570 nm using a BioTek EL ×800 microplate reader (BioTek, Winooski, VT, USA); the value was proportional to the number of viable cells. The viability was calculated as follows: Viability [%] = (A/AC) × 100%; where A is the absorbance of an investigated sample, and AC is the absorbance of control (untreated cells).

For the selection of proper concentrations for tunicamycin and antidepressants used in all experiments, preliminary tests assessing cell viability were performed for concentrations ranging from 0.5–100 µg/mL and 0.1–10 µM for tunicamycin and antidepressants, respectively.

### 2.4. Gene Expression Analysis

Astrocytes were seeded onto culture flasks at a density of 1 × 10^6^ cells/flask and cultured for 24 h. Following this, the culture was treated with tunicamycin (0.5 µg/mL) or tunicamycin with an antidepressant drug (10 µM) for another 24 h. After the incubation, cells were washed with D-PBS, trypsinized and centrifuged at 150× *g*, 5 min, room temperature (RT). The cell pellets were then resuspended in RLT lysis buffer and subjected to RNA isolation and column purification using the NucleoSpin RNA kit. The RNA concentration in the samples was measured with the NanoVue Plus spectrophotometer GE Healthcare (Little Chalfont, UK), and the RNA integrity was determined by electrophoresis using the ScreenTape 2200 system (Agilent Technologies, Santa Clara, CA, USA).

All samples taken for further processing were characterized by RNA Integrity Number (RIN) above 9. A reverse transcription reaction was performed for 1 µg RNA in SureCycler 8800 thermocycler (Agilent Technologies, Santa Clara, CA, USA) using the iScript ™ cDNA Synthesis Kit containing reverse transcriptase and a blend of oligo(dT) and random hexamer primers. cDNA of each sample with a concentration of 10 ng/µL, the 2xSsoAdvanced Universal SYBR Green Supermix reagent, containing dNTPs mixture, thermostable polymerase and SYBR Green marker, were added into appropriate wells on the 96-well defined Custom PrimePCR™ Real-Time PCR Plate according to the manufacturer’s protocol.

All experiments included a PCR reaction control (Prime PCR Control Assay) and RT reaction control (Prime PCR RT Control). The real-time PCR reaction was carried out in a CFX96 thermal cycler (Bio-Rad, Berkeley, CA, USA) and the results were analyzed with the 2-ΔΔCq method. The reference genes GAPDH and TBP were used to normalize the results. Statistical significance was assessed with one-way ANOVA. Fold change value above 2 was chosen as the cut-off criterion. Statistical significance was assumed at *p* < 0.05.

### 2.5. Annexin V Labelling

Apoptosis was detected with the Annexin V-FITC Fluorescence Microscopy Kit. The assay is based on interaction of Annexin V labelled with FITC fluorochrome with phosphatidylserine (PS), exposed during the early stages of apoptosis. For the experiments, cells were seeded onto a 96-well plate to a final density of 8 × 10^3^ cells/well. After 24 h of culture, the cells were exposed to tunicamycin (0.5 µg/mL) or tunicamycin with an antidepressant drug (10 µM) for another 24 h.

After incubation, cells were rinsed with D-PBS and stained with Annexin V-FITC according to the manufacturer’s protocol. Additionally, 1 µg/mL Hoechst 33342 solution was used to stain the cell nuclei. Labelled cells were visualized with the use of InCell Analyzer 2000 (GE Healthcare Life Sciences, Little Chalfont, UK) and analyzed by dedicated InCell Development software. Apoptotic cells were identified on the basis of the fluorescence observed for the FITC channel, while the total numbers of cells, both live and apoptotic, were estimated according to Hoechst staining.

### 2.6. CHOP Protein Expression Analysis

CHOP protein expression was measured using immunoblotting. For the experiments, cells were seeded onto culture flasks to a final density of 1 × 10^6^ cells. After 24 h of culture, the cells were exposed to tunicamycin (0.5 µg/mL) with or without an antidepressant drug (10 µM) for another 24 h. After treatments, the cells were washed in PBS, trypsinized and centrifuged at 150× *g* for 5 min at RT.

Cell pellets were resuspended in lysis buffer (7 M urea, 2 M thiourea, 4% *w/v* CHAPS, 2% *v/v* IPG buffer, 40 mM DTT), sonicated (5 times for 30 s with a 30-s interval, at 4 °C) and centrifuged at 13,000× *g* for 15 min at 4 °C. Concentrations of proteins were measured by 2D Quant kit according to the manufacturer’s instructions. To remove all interfering compounds, a clean-up procedure was performed with the 2D Clean Up Kit according to the manufacturer’s protocol. Electrophoresis and western blotting were performed using stain-free technology Bio-Rad (Berkeley, CA, USA).

Denatured samples (30 μg protein per lane) were separated on Any-kD Mini-Protean TGX stain-free precast gels. Sample integrity and separation quality after electrophoresis was verified with Gel_Doc XR+ Imager (Bio-Rad, Berkeley, CA, USA). The proteins were then transferred to nitrocellulose membranes using the Trans-blot Turbo system. Transfer efficiency was determined using a Gel_Doc XR+ Imager on post-transfer gels and blots.

Thereafter, the membranes were blocked in 3% (*w/v*) non-fat dry milk in TBST (0.1% Tween-20 in Tris Buffered Saline, TBS), for one hour at RT. After overnight incubation at 4 °C with the primary antibody (Anti-DDIT3, 1:250, ab 11419, Abcam, Cambridge, MA, USA), membranes were washed several times in TBST, incubated with secondary antibody at RT for one hour (anti-mouse IgG (HRP), 1:4000, ab205719, Abcam) and washed as described before. Chemiluminescence was performed using Clarity Max Western ECL Substrate kit and detected by ChemiDoc MP imaging system (Bio-Rad, Berkeley, CA, USA). Data were analysed by Image Lab (version 5.2.1). The results were normalized to total protein level according to the manufacturer’s protocol (Stain-free western-blotting protocol, Bio-Rad, Berkeley, CA, USA).

### 2.7. ELISA Assay

The cells were seeded onto culture flasks to a final density of 1 × 10^6^ cells. After 24 h of culture, the cells were exposed to tunicamycin (0.5 µg/mL) with or without R- and S-ketamine (10 µM) for the next 24 h. After treatments, culture medium was collected and centrifuged at 250× *g*, 5 min. BDNF protein levels in the cell-conditioned media were determined using a BDNF ELISA kit, according to the manufacturer’s instructions (R&D Systems, Minneapolis, MN, USA). Data are represented as pg/mL protein.

### 2.8. Data Analysis

Data are expressed as the mean ± standard error of the mean (SEM). The results were tested by one-way ANOVA followed by post hoc Turkey’s multiple comparisons test. All calculations were performed using GraphPad InStat version 9.3.0 (GraphPad, San Diego, CA, USA).

## 3. Results

### 3.1. Effect of Tunicamycin and Antidepressants on the Viability of Astrocytes

Tunicamycin is an antibiotic that is commonly used to induce ER stress and activate the proapoptotic mechanism in the UPR pathway [37]. The present study investigated the effect of tunicamycin on the viability of astrocytes during a 24-h incubation. It was found that the ER stress-inducing agent caused a significant and dose-dependent reduction in cell viability (Figure 2A). In further experiments, to avoid the cytotoxic effects of the drug, a concentration of 0.5 µg/mL was chosen; at this concentration tunicamycin evoked a significant, but not excessive, decrease in cell viability, i.e., up to 70% compared to control values.

In contrast, the tested antidepressants did not induce changes in cell viability at concentrations of 0.1 and 1 µM; however, at a concentration of 10 µM, escitalopram, R-ketamine and S-ketamine evoked a statistically significant increase in formazan accumulation, i.e., 16%, 15% and 17% compared to controls respectively (Figure 2B). Therefore, this concentration (10 µM) of antidepressants was selected for further studies.

### 3.2. Effects of Tunicamycin and Antidepressants on the Viability of Astrocytes Undergoing ER Stress

Concomitant 24-h treatment of astrocytes with 10 µM antidepressants and tunicamycin (0.5 µg/mL) caused a decrease in MTT conversion: formazan accumulation was found to be 74% of control values for both amitriptyline and escitalopram, 67% for R-ketamine and 72% for S-ketamine. In comparison, the viability of tunicamycin-treated cells was decreased to 70% of control values (Figure 3). The results showed that antidepressant drugs were not able to reversed/counteract the tunicamycin effect on cell viability.

### 3.3. Effects of Antidepressants on Gene Expression Related to ER Stress in Astrocytes

To investigate the effect of antidepressants on gene expression in astrocytes following ER stress, the most characteristic genes in the classic UPR pathway were selected for the study (Table 1). Most ER stress-responsive genes demonstrated significantly greater expression following 24-h incubation with 0.5 µg/mL tunicamycin (Figure 4; Appendix A). Among them, HSPA5 and DITT3 were significantly upregulated, with fold change of 19.4 and 13.7, respectively, while CREB3L4, EDEM1 and ERN1 gene expression increased by 3.9-, 3.8- and 3.6-fold, respectively. ATF4, ATF6 and CREB3 showed only moderate changes in expression, slightly exceeding a 2-fold increase (i.e., 2.9, 2.5 and 2.3-fold change values, respectively) (Figure 4; Appendix A).

It was found that 24-h incubation with the tested antidepressants (10 µM) did not significantly affect the expression of selected ER stress-responsive genes: nearly all fold changes were below 2—while ERN-1 demonstrated a 2.1-fold increase under the influence of R-ketamine and 2.2-fold under S- ketamine. Additionally, R-ketamine slightly changed CREB3L1 gene expression, showing a 2.1-fold increase (Table 1).

To show the difference in gene expression in astrocytes in response to ER stress affected or not by antidepressants, the symbol delta (Δ) was introduce. The symbol Δ means the difference between the gene expression levels changed by antidepressants in tunicamycin-treated astrocytes versus cells exposed to tunicamycin alone. Addition of amitriptyline and escitalopram (10 µM) for 24 h to the tunicamycin-treated cells increased only the expression of DITT3 and CREB3L4 compared with tunicamycin alone (Δ 2.8-fold and 1.6-fold, respectively) (Figure 4; Appendix A).

Following 24-h incubation with R-ketamine, the tunicamycin-treated astrocytes demonstrated a high and statistically significant increase in the expression of DITT3 (Δ fold change was 4.6), and to a lesser extent ERN1, CREB3L1 and EDEM1 genes (Δ fold change values were 2.8, 1.5 and 1.2, respectively).

S-ketamine treatment appeared to have a marked influence on the expression of genes related to ER stress in astrocytes. The addition of 10 µM of S-ketamine for 24 h significantly elevated levels of DITT3 and ERN1 mRNA in tunicamycin-treated cells compared with the cells treated with tunicamycin alone (Δ fold change values were 5.8 and 4.6, respectively). S-ketamine also activated all tested OASIS family genes (CREB3L4, CREB3 and CREB3L1) in astrocytes exposed to ER stress compared to tunicamycin alone (Δ fold change values were 2.2, 1.0, 1.7, respectively), and to a lesser extent, EDEM1 (Δ fold change of 1.7) (Figure 4; Appendix A).

### 3.4. Apoptosis Induction

As R-ketamine and S-ketamine induced more genes associated with ER stress response than amitriptyline and escitalopram (Table 1; Figure 4; Appendix A), the two ketamine enantiomers were subjected to further analysis. Moreover, as R-ketamine and S-ketamine effectively and in a statistically significant manner increased expression of DDIT3, the gene involved in proapoptotic mechanisms of the UPR pathway; therefore, their effects on the apoptosis of astrocytes were also checked. After 24 h, a significant rise in the number of apoptotic cells was observed in tunicamycin-treated cells. Although both S-ketamine and R-ketamine greatly increased DDIT3 gene expression in the tunicamycin-treated astrocytes, neither drug influences tunicamycin-induced apoptosis, as measured by annexin V labelling: the percentage of apoptotic cells remained at the level of untreated cells (Figure 5).

### 3.5. CHOP Protein Expression

Due to important upregulation of the DDIT3 gene induced by tunicamycin and the fact that the greatest and statistically significant difference (Δ) in DDIT3 gene expression was induced by S-ketamine in tunicamycin-treated astrocytes (Figure 4; Appendix A), the next part of the study examined its influence on the CHOP protein encoded by the DDIT3/CHOP gene. In contrast to the significantly increased DDIT3 expression by S-ketamine in astrocytes undergoing tunicamycin-induced ER stress, the upregulation of CHOP protein was even diminished (508% vs. 339% of control for tunicamycin and S-ketamine with tunicamycin, respectively) (Figure 6). Furthermore, S-ketamine had no statistically significant effect on CHOP protein expression when given alone to astrocyte culture (Figure 6).

### 3.6. BDNF Release

Considering that BDNF expression is reduced in the pathophysiology of depression, the study examined the effects of R-ketamine and S-ketamine treatment on astrocytes under normal conditions and tunicamycin-induced ER stress. S-ketamine in a statistically significant manner increased the release of BDNF in astrocytes not treated by tunicamycin. In contrast, R-ketamine treatment yielded a smaller increase that was not statistically significant. Neither R- or S-ketamine had not statistically significant influence on the tunicamycin-induced decrease in BDNF release in astrocytes (Figure 7).

## 4. Discussion

Despite the high prevalence of depression worldwide there is still no effective treatment, particularly considering the adverse effects of antidepressants, the delay to onset of full effectiveness and the relapse rate. Therefore, there is an urgent need to identify new targets for the treatment of depression. One such target could be the UPR pathway. As the effect of existing antidepressant drugs on the UPR pathway is not well understood the present study examines the influence of selected antidepressants on the expression of ER stress-responsive genes in a human astrocyte model.

Astrocytes are an abundant subgroup of cells in the CNS that play a critical role in controlling neuronal circuits involved in emotion, learning and memory [38]. Post-mortem studies on human brain tissues of patients with mood disorders show consistent changes in glial cell morphology, as well as reductions in astrocyte density, with a concomitant decrease in astrocyte-related biomarkers and genes, indicating that astrocytes may play a role in psychiatric conditions [28,29,38]. Hence, in the present study, a human astrocyte cell line was exposed to the ER stress-inducing agent tunicamycin. This compound mimics the processes taking place in the development of depression, i.e., by inhibiting protein glycosylation, resulting in accumulation of misfolded proteins in astrocytes and decreasing the release of BDNF in astrocytes (Figure 4 and Figure 7; Appendix A).

In the present paper, 0.5 µg/mL of tunicamycin was found to evoke a 70% reduction in astrocyte viability, accompanied by a moderate increase in the percentage of apoptotic cells. To confirm that tunicamycin (0.5 µg/mL) is a reliable ER stress inducer in human astrocyte cell line, we analysed changes in the expression of genes involved in the UPR pathway. Tunicamycin significantly (i.e., fold change above 2), increased the expression of most genes selected.

Therefore, the next stage of the study compared the effects of three different antidepressants, i.e., amitriptyline (a TCA), escitalopram (an SSRI) and S-ketamine (NMDA receptor antagonist—newly approved by FDA), on the genes of the UPR pathway following ER stress. In addition, R-ketamine was also included in the analysis. R-ketamine is a less active enantiomer of S-ketamine and is currently under evaluation in clinical trials. Based on literature data and our own cell viability measurements, a 10 µM concentration of antidepressants was chosen for the analysis [35,39,40].

In the present study, escitalopram, was not found to alter, in a statistically significant manner, the expression of genes of the UPR pathway, neither under normal conditions nor following tunicamycin-induced ER stress. The SSRI group demonstrates different effects, sometimes opposite, on the UPR pathway. For example, sertaline was found to induce ER stress in hepatic cells, leading to hepatotoxicity; this was attributed to two UPR branches, viz. PERK-eIF2α-ATF4 and IRE1-XBP-1, leading to an increase of CHOP expression at the mRNA and protein levels [41]. In contrast, fluoxetine inhibited ER stress in a rat model by suppressing the expression of mRNA and protein of GRP78 and CHOP [36].

However, the same drug increased CHOP mRNA and protein expression upon activation of PERK-eIF2α-ATF4 and ATF6 cascade in glioma cells [35]. The administration of escitalopram led to a reduction of GRP78 and CHOP protein expression and decreased the protein level of caspase-12 in the hippocampal nerve cells of chronic unpredictable mild stress (CUMS)-exposed rats, i.e., an animal model of depression [42]. Escitalopram has also been found to exert an antidepressant effect by attenuating tunicamycin-induced ER stress in brain microvascular endothelial cells by reducing the expression of PERK, GRP78, XBP1 and CHOP proteins [43].

In our studies, surprisingly, escitalopram did not change the expression of DDIT3/CHOP or GRP78 genes following tunicamycin-induced ER stress in astrocytes, but increased the expression of CREB3L4, which encodes a protein belonging to OASIS family members (Figure 4; Appendix A). The OASIS family members i.e., OASIS, CREB4 and Luman, manage aspects of biological regulation, including cell differentiation, maturation and basal cellular homeostasis, and they may be activated as transcription factors in response to the ER stress [44]. However, further research is needed to confirm whether the escitalopram-induced expression of CREB3L4 gene might be implicated in mechanism of the UPR pathway activation in depression.

Another tested antidepressant drug was the TCA amitriptyline. Twenty-four-hour incubation of astrocytes with amitriptyline alone did not appear to influence the expression of the tested UPR pathway genes. However, treatment of astrocytes with amitriptyline under tunicamycin-induced ER stress markedly increased DDIT3/CHOP mRNA expression (Δ fold change was 2.83) compared to the cells treated with tunicamycin alone (Figure 4; Appendix A). These results are in line with previous findings about TCAs: Ma et al. [39] report that the CHOP-dependent ER stress pathway is responsible for desipramine-induced apoptosis in C6 glioma cells, and it appears that this apoptotic effect occurs through the PERK-eIF2α and ATF6 signalling pathways [45].

Approximately one third of depressed patients fail to respond to currently-available antidepressant therapies [46]. Therefore, new drugs are being sought, while existing drugs, such as ketamine, are being repurposed for novel therapies. For many years, ketamine has been successfully used as a general anaesthetic; however, over the past two decades, it has been found to offer promise for treating drug-resistant depression (TRD). Ketamine has also been found to significantly reduce the frequency of suicidal thoughts in depressive patients [47] and unlike other existing antidepressants, it demonstrates a rapid onset of action and long-lasting effects.

The well-known mechanism of action of ketamine is the blockade of glutamate receptors [48]. However, glutamate-independent effects of the drug has been also highlighted, such as an influence on monoaminergic system (dopamine, serotonin, noradrenaline, GABA), neurotrophins (like BDNF, vascular endothelial growth factor—VEGF), or mammalian target of rapamycin (mTOR) and UPR signalling pathways [49,50,51,52,53]. Ketamine exists in two types of enantiomers: S-ketamine and R-ketamine. Studies suggest that R-ketamine has stronger and longer lasting antidepressant effects than S-ketamine [54]; however, only S-ketamine has so far been approved by the FDA for use as an antidepressant drug, and clinical trials of R-ketamine are currently underway [55,56].

There is little information about the mechanism of action of ketamine on the UPR pathway in the brain. Studies on rats suggest that it might have an influence on levels of PERK, IRE1 and CHOP protein and that its activity may be dependent on the region of the brain; there may also be an interaction between the UPR and mTOR pathways, and it has been proposed that ketamine may exert its quick action due to the activation of mTOR signalling [50]. Ketamine was found to activate the expression of ER stress-related proteins in adult neural stem cells; however, much higher levels of ketamine (400 µM) were needed to elicit the observed increase of CHOP mRNA and protein than were used in this study or in clinical practice [57].

In the periphery ketamine also induced cystitis (KC) due to the upregulation of GRP78 and CHOP production, resulting in apoptosis in rats and SV-HUC-1 human uroepithelial cells [58]. Other ketamine-associated cystitis (KC) models obtained similar results; however, the toxic effect of ketamine was found to involve another signalling cascade resulting in cell apoptosis: the IRE1-TRAF2-ASK1-JNK pathway [40]. Interestingly, no upregulation of the MAP3K4 gene encoding the protein activating the JNK pathway was observed in the present study by any of the used antidepressants.

In this paper, S-ketamine did not appear to have any significant influence on the expression of most genes of the UPR pathway in astrocytes under normal conditions (except for the ERN1/IRE1 gene); however, it significantly increased the expression of most tested genes in astrocytes undergoing tunicamycin-induced ER stress (Figure 4; Appendix A) with the greatest effect observed for the DDIT3/CHOP gene. It is important to mention that the DDIT3/CHOP gene encodes CHOP protein, a transcription factor involved in induction of apoptosis when the ER function cannot be restored. Surprisingly, S-ketamine did not affect tunicamycin-induced apoptosis or the expression of CHOP protein, which was even slightly diminished. It can be concluded that S-ketamine-induced high expression of DDIT3/CHOP on gene level does not correspond with the protein expression.

Interestingly, the UPR pathway is not only involved in toxic effects but it may also play beneficial role in maintaining neuronal activities and function under normal condition: the IRE1-XBP1 signalling pathway of the UPR was found to be involved in the expression of BDNF in neurons, resulting in promotion of dendritic elongation and branching [59]. Moreover, exogenous treatment of BDNF also drives the expression of neurotrophin via IRE1-XBP1 signalling cascades.

This implies that the BDNF released by astrocytes may have an impact on neuronal functioning. In our studies, S-ketamine and, to a lesser degree, R-ketamine increased BDNF release in astrocytes, which may suggest that ketamine has a glutamate-independent mechanism of action, possibly involving the up-regulation of ERN1/IRE1 mRNA (Figure 1 and Figure 7; Table 1).

It is important to note that S-ketamine also upregulated genes encoding the OASIS family members (CREB3, CREB3L1 and CREB3L4), which share a region with high sequence similarity to ATF6 and are expressed preferentially in astrocytes. To date, no studies have been conducted to demonstrate the relationship between OASIS and S-ketamine, but considering that ER stress promotes the expression of OASIS family members, which have a protective function, further studies are needed to explain whether the mechanism of S-ketamine action is related to other signalling pathways originating from the ER [60].

R-ketamine was found to exert greater potency and longer-lasting antidepressant effects than S-ketamine in a rodent model of depression [54]. Therefore, the present study compared the effect of the two enantiomers on the expression of ER stress-responsive genes in human astrocyte cell line. R-ketamine was found to exert a similar influence on the expression of the UPR pathway genes as S-ketamine but the effect was milder. Interestingly, only R-ketamine increased the expression of CREB3L1, encoding OASIS in astrocytes, under normal conditions (Table 1), while it also enhanced CREB3/LUMAN mRNA expression in astrocytes under ER stress; this supports the idea of that both ketamine enantiomers may interact with OASIS family members.

## 5. Conclusions

In conclusion, depression remains a growing therapeutic problem. Aside from the attempted suicide of ineffectively-treated patients, many treatments demonstrate delayed curative effects, adverse effects and resistance to traditional antidepressants. There is hence a need to identify new therapeutic options for MDD treatment. One promising novel drug target is associated with the activation of the UPR pathway involved in the course of depression. Our present findings indicate that only S-ketamine, and R- ketamine to a lesser extent, significantly activated the most characteristic genes of the classic UPR pathway in astrocytes under ER stress.

Furthermore, cell viability and apoptosis measuring assays showed that S-ketamine did not appear to affect astrocyte survival under ER stress conditions. Interestingly, under unstressed conditions, both S-ketamine and R-ketamine (again, to a lesser extent) increased the release of BDNF, indicating that the effects of the drug show a complex mechanism of action in astrocytes. Our results are the first to show the possible influence of S-ketamine on stabilizing the expression of OASIS family members following ER stress; however, the potential role of OASIS in the mechanism of therapeutic action of ketamine requires further study.

## Figures and Tables

**Figure 1 pharmaceutics-14-00846-f001:**
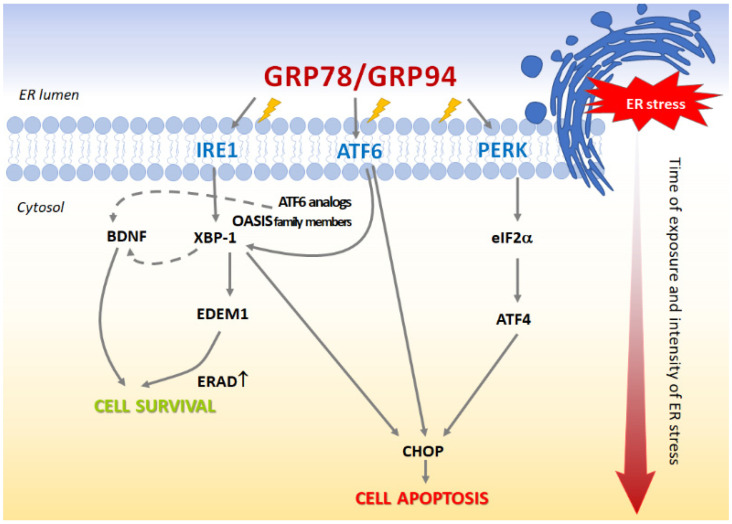
Under ER stress, the UPR pathway is activated to minimize the accumulation of unfolded and misfolded proteins in the ER. ATF4, *activating transcription factor 4*; ATF6, *activating transcription factor 6*; BDNF, *brain-derived neurotrophin factor*; CHOP, *C/EBP-homologous protein*; EDEM1, ER *Degradation-Enhancing Alpha-Mannosidase-Like Protein 1*; eIF2α, *eukaryotic translation initiation factor 2α*; ER, *endoplasmic reticulum*; ERAD, *endoplasmic reticulum associated degradation*; GRP78, *78-kDa glucose-regulated protein*; GRP94, *94-kDa glucose-regulated protein*; IRE1, *inositol requiring enzyme 1*; XBP1, *X-box binding protein 1*; OASIS, *old astrocyte specifically induced substance*; PERK, *protein kinase R-like ER kinase*; dashed line—indirectly involved in the UPR pathway.

**Figure 2 pharmaceutics-14-00846-f002:**
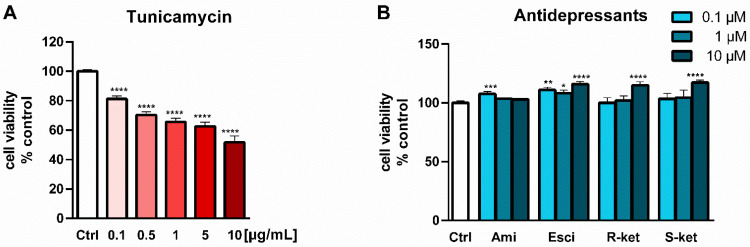
Designing an experimental model for further research work: Effect of tunicamycin (0.1–10 µg/mL) on the viability of astrocytes (**A**). Effects of amitriptyline (Ami), escitalopram (Esci), R-ketamine (R-ket) and S-ketamine (S-ket) on the viability of astrocytes after 24-h incubation (**B**). Data are presented as the mean ± SEM and expressed as a percentage of untreated control cells. Statistical significance vs. control cells is indicated when appropriate; * *p* < 0.05; ** *p* < 0.01; *** *p* < 0.001; **** *p* < 0.0001.

**Figure 3 pharmaceutics-14-00846-f003:**
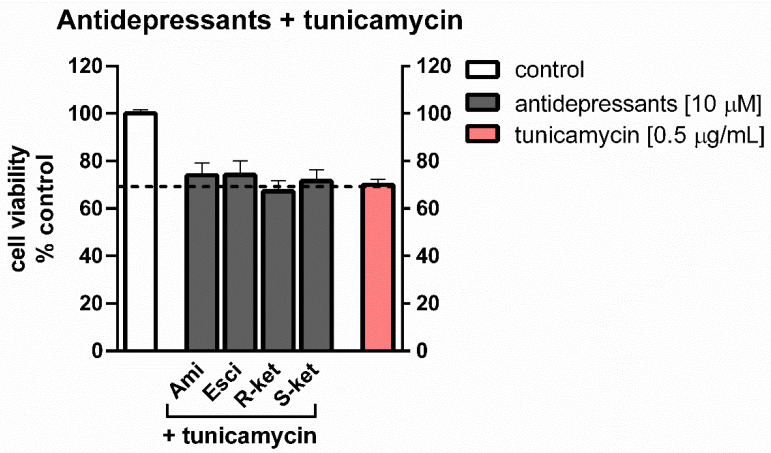
Effects of amitriptyline (Ami), escitalopram (Esci), R-ketamine (R-ket) and S-ketamine (S-ket) on viability of astrocytes during 24-h tunicamycin-induced ER stress. Data are presented as the mean ± SEM and expressed as a percentage of untreated control cells.

**Figure 4 pharmaceutics-14-00846-f004:**
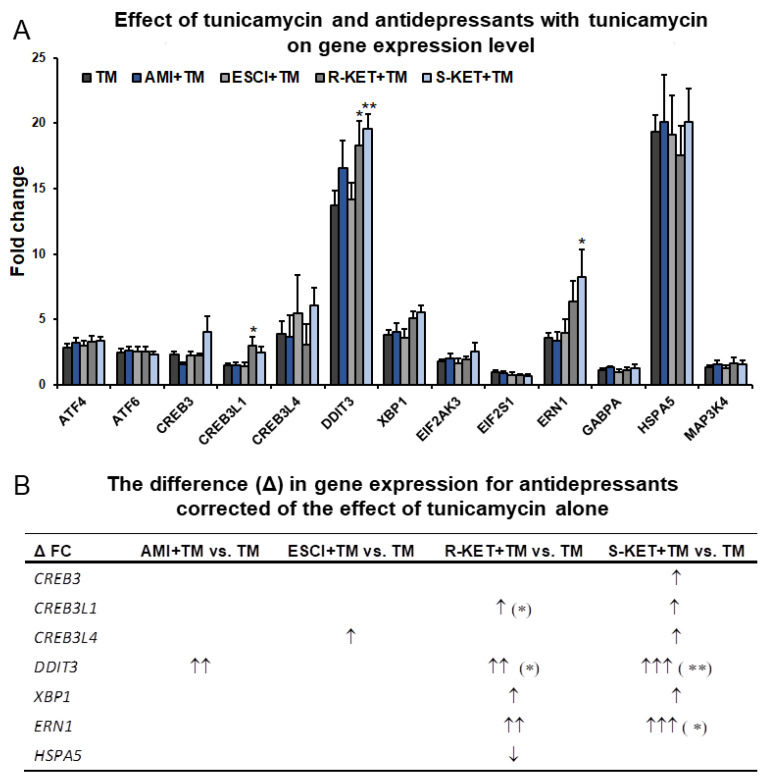
(**A**) Effect of tunicamycin (TM) with antidepressants (Ami—amitryptyline, Esci—escitalopram, R-ket—R-ketamine and S-ket—S-ketamine) and tunicamycin alone in astrocytes with regard to the expression of genes associated with ER stress. (**B**) To illustrate the difference (Δ) in gene expression of antidepressants the symbols were introduced: ↑ for 1–2.5 fold change; ↑↑ 2.6–4.5 fold change; ↑↑↑ > 4.6 fold change. FC—fold change. Data are presented as the mean ± SEM and expressed as fold change vs. untreated control cells. Statistical significance vs. tunicamycin-treated cells is indicated when appropriate; * *p* < 0.05; ** *p* < 0.01.

**Figure 5 pharmaceutics-14-00846-f005:**
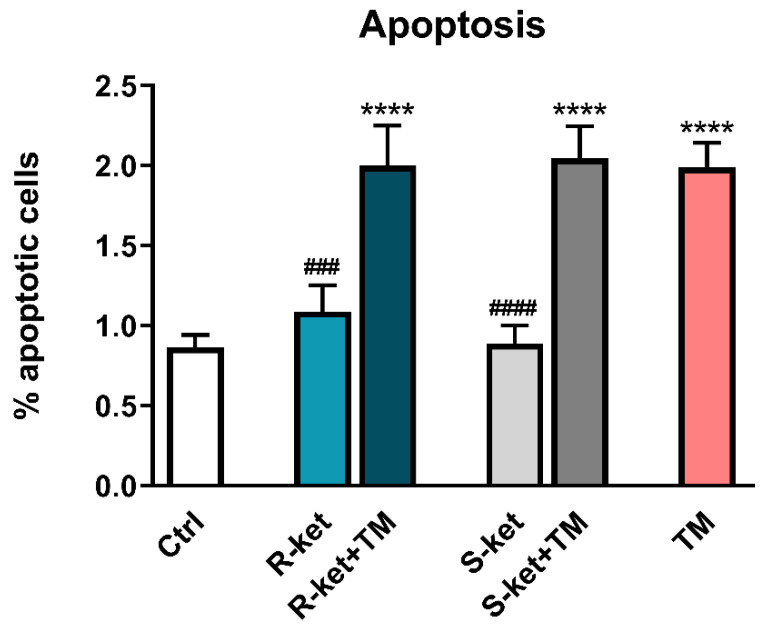
Effects of tunicamycin alone (TM), R-ketamine alone (R-ket), S-ketamine alone (S-ket) or R-ketamine and S-ketamine in tunicamycin-treated astrocytes (R-ket + TM or S-ket + TM) on induction of apoptosis in astrocytes. Data are presented as the mean ± SEM and expressed as a percentage of untreated control cells. Statistical significance vs. control cells is indicated when appropriate; **** *p* < 0.0001 vs. untreated control; ### *p* < 0.001 and #### *p* < 0.0001 vs. tunicamycin-treated cell.

**Figure 6 pharmaceutics-14-00846-f006:**
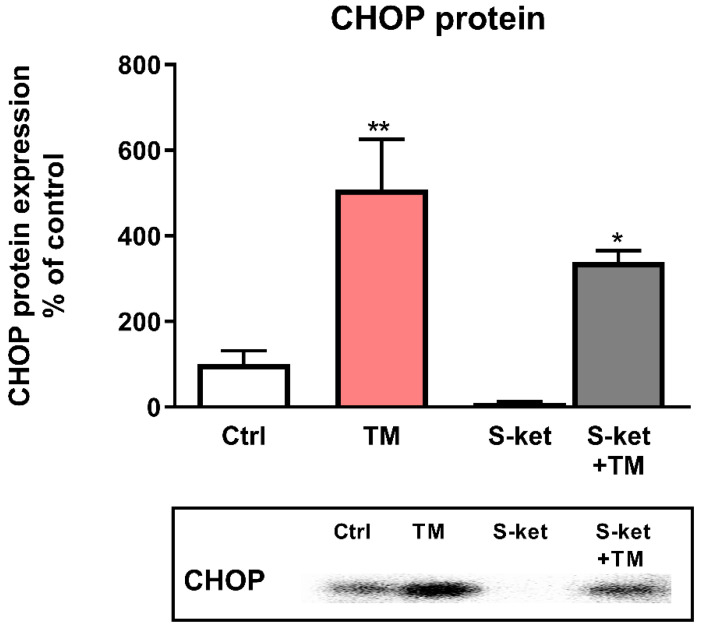
Effects of tunicamycin alone (TM), S-ketamine alone (S-ket) and S-ketamine in tunicamycin-treated astrocytes (S-ket + TM) on CHOP protein expression. Data are presented as the mean ± SEM and expressed as a percentage of untreated control cells. Statistical significance vs. control cells is indicated when appropriate; * *p* < 0.05; ** *p* < 0.01.

**Figure 7 pharmaceutics-14-00846-f007:**
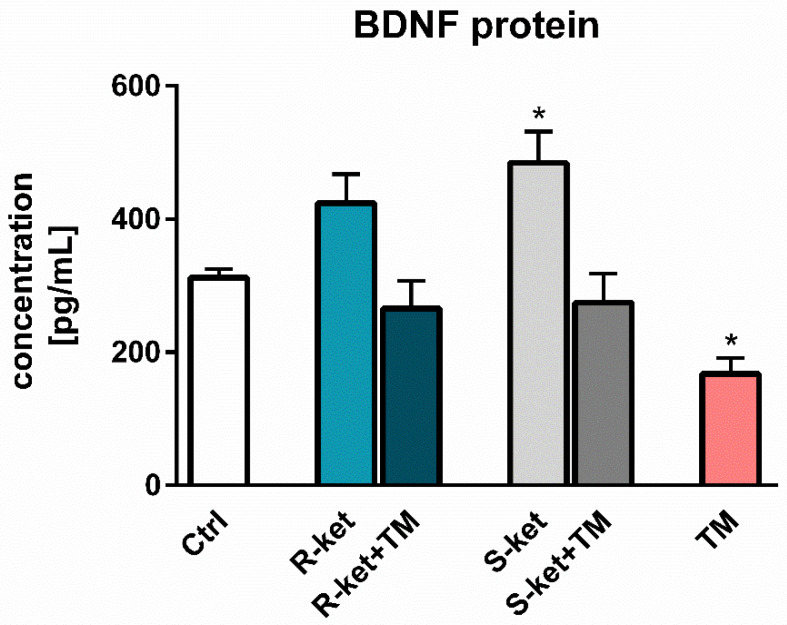
Effects of tunicamycin alone (TM), R-ketamine alone (R-ket), S-ketamine alone (S-ket) or R-ketamine and S-ketamine in tunicamycin-treated astrocytes (R-ket + TM or S-ket + TM) on BDNF protein release. Data are presented as the mean ± SEM and expressed as a percentage of untreated control cells. Statistical significance vs. control cells is indicated when appropriate; * *p* < 0.05.

**Table 1 pharmaceutics-14-00846-t001:** Effects of amitriptyline (Ami), escitalopram (Esci), R-ketamine (R-ket) and S-ketamine (S-ket) on gene expression of ER stress-responsive genes. The fold change values >2 were marked in green. Data are presented as the mean ± SEM and expressed as fold change vs. untreated control cells. –: not studied; * According to https://www.proteinatlas.org/, accessed on 1 February 2022; ^#^ more frequent in usage in the literature.

Studied Gene	Alternative Name	Encoded Protein *	Fold Change			
			Ami	Esci	R-ket	S-ket
* **ATF4** *	*CREB-2*	*Activating transcription factor 4*	0.98 ± 0.08	0.84 ± 0.13	1.10 ± 0.14	0.86 ± 0.14
* **ATF6** *	*-*	*Activating transcription factor 6*	1.01 ± 0.13	0.74 ± 0.05	0.98 ± 0.1	1.00 ± 0.2
* **CREB3** *	*Luman*	*CAMP responsive element binding protein 3*	–	–	0.82 + 0.01	1.19 ± 0.44
* **CREB3L1** *	*Oasis*	*CAMP responsive element binding protein 3 like 1*	1.30 ± 0.13	0.89 ± 0.08	2.11 ± 0.66	1.85 ± 0.5
* **CREB3L4** *	*CREB4*	*CAMP responsive element binding protein 3 like 4*	–	–	1.21 ± 0.39	1.25 ± 0.53
* **DDIT3** *	*CHOP ^#^*	*DNA damage inducible transcript 3/C/EBP-homologous protein*	1.21 ± 0.22	0.99 ± 0.26	1.21 ± 0.12	1.08 ± 0.31
* **EDEM1** *	*EDEM*	*ER degradation enhancing alpha-mannosidase like protein 1*	1.27 ± 0.14	0.86 ± 0.01	1.02 ± 0.15	1.05 ± 0.23
* **EIF2AK3** *	*PERK*	*Eukaryotic translation initiation factor 2 alpha kinase 3*	1.38 ± 0.13	0.91 ± 0.07	1.06 ± 0.21	1.08 ± 0.23
* **EIF2S1** *	*eIF2α*	*Eukaryotic translation initiation factor 2 subunit alpha*	1.17 ± 0.07	1.03 ± 0.00	0.89 ± 0.25	0.74 ± 0.12
* **ERN1** *	*IRE1*	*Endoplasmic reticulum to nucleus signalling 1/Inositol-requiring enzyme 1*	1.38 ± 0.41	0.93 ± 0.07	2.06 ± 0.97	2.22 ± 0.66
* **GABPA** *	*NRF2*	*GA binding protein transcription factor subunit alpha*	1.07 ± 0.07	0.98 ± 0.12	1.03 ± 0.25	0.95 ± 0.19
* **HSPA5** *	*GRP78*	*Heat shock protein family A (Hsp70) member 5*	1.11 ± 0.03	0.98 ± 0.05	0.90 ± 0.08	0.86 ± 0.19
* **MAP3K4** *	*MEKK4*	*Mitogen-activated protein kinase kinase kinase 4*	1.13 ± 0.14	0.89 ± 0.07	1.24 ± 0.71	1.09 ± 0.25

## Data Availability

Not applicable.

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
