# Peer review of "The Importance of Endoplasmic Reticulum Stress as a Novel Antidepressant Drug Target and Its Potential Impact on CNS Disorders"

_pharmaceutics, 2022, doi:10.3390/pharmaceutics14040846_

Round 1

Reviewer 1 Report

The importance of endoplasmic reticulum stress as a novel antidepressant drug target and its potential impact on CNS disorders.

By Marta Jóźwiak-Bębenista et al.

The paper is devoted to the nature of depression and the search for new antidepressants. The problem is relevant, since the causes of depression are unknown, and the mechanism of action of the drugs used, such as ketamine, is also unknown.

It is known that depression causes EP stress and accumulation of unfolded and misfolded proteins in the ER. It is also known that there is a reductions in astrocyte density in patients with mood disorders and  changes in cell morphology, resulting in accumulation of misfolded proteins, and decreasing the release of BDNF in astrocytes.

Therefore, the aim of the work was to test how antidepressants (ketamines) affect the expressions of genes related to the UPR signaling pathways on astrocytes. An antibiotic Tunicamycin was used to activate ER stress. The results showed that antidepressant drugs were not able to reversed/counteract the tunicamycin effect on cell viability. Authors concluded that astrocytes are not involved in the UPR path way-dependent cytotoxic effects of ketamines in the brain. Although some new possible mechanisms of the UPR pathway activation in depression are shown. Thus, a negative result was obtained in the work. The article presents well-described reliable experiments. The article can be accepted as presented.

Author Response

Reviewer # 1

We would like to thank you for careful reading of our manuscript and positive response. We appreciate the positive feedback from the Reviewer.

Reviewer 2 Report

The study, focused on the analysis of changes in endoplasmic reticulum stress under treatment with antidepressants, is novel and could raise some interest in the journal readers. However, the project design conceals significant conceptual limitations.

What is the rational for the use of astrocytes in culture as a model? No mention about the reason(s) of this choice was presented in the introduction.

This is especially relevant in consideration of the fact that monoaminergic drugs are supposed to act modulating monoaminergic transmission (as recognised also by the authors). How the authors suppose these drugs may directly act on astrocytes?

The use of monoaminergic drugs on astrocytes makes absolutely no sense in my view. Thus, I would strongly suggest to limit the study on the effects of (R), (S) ketamine and to rather consider to test the effects of (R)/(S)-ketamine.  

For this reason, I recommend rejection with resubmission.

Please also note that the model used is not “primary human astrocytes” as stated in the introduction, but an astrocyte cell line.

Other minor comments are as follows:

Abstract:

The sentence “Furthermore, cell viability and apoptosis measuring assays showed that (R-)S-ketamine did not affect cell survival under ER stress; this suggests that astrocytes are not involved in the UPR pathway dependent cytotoxic effects of S-ketamine in the brain” is unclear.

The sentence “S-ketamine played the key role in increasing the release of brain-derived neurotrophic factor (BDNF), indicating that the drug has various glutamate-independent therapeutic effects.” makes no sense.  Changes in glutamate transmission are correlated to BDNF release.

Introduction:

Lines 53-60 were embedded in the ms text but should be part of Figure 1 legend

Lines 63-64 “It has also been shown that ER stress is also associated with neurodegenerative diseases

Lines 76-87: theories of depression and drug treatment are not properly presented. The most widely recognized theory is based on neuroplasticity and must be adequately discussed as an evolution of the monoaminergic hypothesis. Confusion must be avoided. Moreover, ketamine can not be put on the same level as TCA!! Racemic ketamine has been introduced as off-label treatment for TRD!

Lines 90-91: The study from Bown et al. seems to highlight a role for alterations in the UPR pathway in association with suicide more than with depression. I would instead give more space to previous preclinical evidence.

Results:

Reporting all the qPCR results in tables does not help in their understanding. I would suggest to consider to graph at least the significant ones.

Author Response

Reviewer # 2

We would like to thank the Reviewer for careful and thorough reading of the manuscript and for the thoughtful comments and constructive suggestions, which help to improve the quality of our manuscript. We highlighted in yellow the changes in the revised manuscript according to the comments of Reviewer # 2.

The study, focused on the analysis of changes in endoplasmic reticulum stress under treatment with antidepressants, is novel and could raise some interest in the journal readers. However, the project design conceals significant conceptual limitations.

We have carefully read all comments and revised the manuscript as per the Reviewer’s suggestions. We hope the revised manuscript will meet the Reviewer’s expectations.

What is the rational for the use of astrocytes in culture as a model? No mention about the reason(s) of this choice was presented in the introduction.

We agree with the Reviewer’s suggestion. We have included the reasons in the discussion section but it was not enough highlighted. The authors expanded this part and explained according to Reviewer’s suggestion in the introduction section why human astrocyte cell line was used as a model in our studies.

This is especially relevant in consideration of the fact that monoaminergic drugs are supposed to act modulating monoaminergic transmission (as recognised also by the authors). How the authors suppose these drugs may directly act on astrocytes?

It is important to note that astrocytes express transporters for both norepinephrine (NET) and serotonin (SERT), which are the targets of several classical antidepressant drugs. This raises a possibility that antidepressants can have direct effects on astrocytes by blocking the reuptake of monoamines by astrocytes. Moreover, astrocytes abundantly express α2A and β1 adrenoreceptors, with α1A expressed at much lower levels. Astrocytes also express 5-HT1A, 5-HT2A, and 5-HT2B receptors of serotonin, in addition to 5-HT5A, which is a predominantly astrocyte-specific receptor. Several studies have revealed that these receptors respond to physiologically relevant stimuli such as calcium influx and cyclic adenosine monophosphate (cAMP) concentrations, suggesting that these receptors could play an important role in antidepressant-mediated changes in monoamine concentrations [Marathe et al., 2018]

Marathe, S.V.; D’Almeida, P.L.; Virmani, G.; Bathini, P.; Alberi, L. Effects of Monoamines and Antidepressants on Astrocyte Physiology: Implications for Monoamine Hypothesis of Depression. J. Exp. Neurosci. 2018, 12, 1179069518789149.

The use of monoaminergic drugs on astrocytes makes absolutely no sense in my view. Thus, I would strongly suggest to limit the study on the effects of (R), (S) ketamine and to rather consider to test the effects of (R)/(S)-ketamine.  

For this reason, I recommend rejection with resubmission.

The goal of the study was the examine whether, and to what extent, commonly used antidepressant drugs and two types of enantiomers of ketamine: S-ketamine and R-ketamine affect the expressions of genes related to the UPR signalling pathways in human astrocyte cell line because ER stress in astrocytes may have consequences in the functioning of neurons (refer to the introduction section where a justification of the use of astrocytes in culture was included, please). Moreover, according to literature data, the effects of monoaminergic drugs on genes of the UPR pathway have been studied as a new possible mechanism of their action, but the results remain not clear and varies from researchers. Especially the SSRI group demonstrates different effects, sometimes opposite, on the UPR pathway, therefore we thought it might be interesting to widen our studies including commonly used antidepressant drugs. We observed that amitriptyline increased mRNA CHOP expression in tunicamycin-induced ER stress in astrocyte cell line which might indicate possible involvement of the drug in the activation of the UPR pathway. We are aware that the result was not significant but this observation might be valuable and helpful for the researchers studying the same issue (especially that results are often opposite).

For these reasons we would like to ask the Reviewer for the opportunity to present the effects of monoaminergic drugs on the genes of the UPR pathway in our manuscript.

Only S-ketamine, not (R)/(S)-ketamine (racemate) was approved by the FDA for the treatment of refractory depression therefore racemate of ketamine was not interesting in our point of view. The racemate of ketamine in used in anesthesiology.

Studies suggest that R-ketamine has stronger and longer lasting antidepressant effects than S-ketamine [Zhang et. al 2014]; however, only S-ketamine has so far been approved by the FDA for use as an antidepressant drug, and clinical trials of R-ketamine are currently underway [Yang et al., 2015; Mansouri et al., 2017]. Our results have shown that there are slight differences between the isomers. It’s very interesting that racemic ketamine has been introduced as off-label treatment for TRD we take it into consideration in the next step of our project. Thank you for this valuable suggestion.

Zhang, J.C.; Li, S.X.; Hashimoto, K. R (-)-ketamine shows greater potency and longer lasting antidepressant effects than S(+)-ketamine. Pharmacol Biochem Behav. 2014, 116, 137–141.

Yang, C.; Shirayama, Y.; Zhang, J. C.; Ren, Q.; Yao, W.; Ma, M.; Dong, C.; Hashimoto, K. R-ketamine: a rapid-onset and sustained antidepressant without psychotomimetic side effects. Transl psychiatry. 2015, 5, e632.

Mansouri, S.; Agartz, I.; Ögren, S.O.; Patrone, C.; Lundberg, M. PACAP Protects Adult Neural Stem Cells from the Neurotoxic Effect of Ketamine Associated with Decreased Apoptosis, ER Stress and mTOR Pathway Activation. PLoS One. 2017, 12, e 0170496.

Please also note that the model used is not “primary human astrocytes” as stated in the introduction, but an astrocyte cell line.

Thank you for reading the manuscript carefully and noticing the mistakes. It has been corrected

Other minor comments are as follows:

Abstract:

The sentence “Furthermore, cell viability and apoptosis measuring assays showed that (R-)S-ketamine did not affect cell survival under ER stress; this suggests that astrocytes are not involved in the UPR pathway dependent cytotoxic effects of S-ketamine in the brain” is unclear.

The sentence has been changed.

The sentence “S-ketamine played the key role in increasing the release of brain-derived neurotrophic factor (BDNF), indicating that the drug has various glutamate-independent therapeutic effects.” makes no sense.  Changes in glutamate transmission are correlated to BDNF release.

We would like to mention that the release of BDNF by S-ketamine may result in glutamate-dependent as well as glutamate-independent way. Yu et al. reported that a single dose of NMDA receptor antagonist, MK801, induced acute and short-lasting behavioural alterations in rats and the observed effect was connected with a marked, transient increase in BDNF expression [Manahan-Vaughan et al., 2008; Yu et. al. 2021]. However, other mechanism besides glutamate system can also lead to the release of BDNF. The cAMP/PKA pathway has been shown to be involved in antidepressant-induced BDNF expression in astrocytes [Zhou et al., 2019]. The high concentration of monoamines (such as norepinephrine, serotonin and dopamine) induced by monoaminergic drugs, initiated cAMP-dependent release of BDNF in astrocytes. Additionally, the IRE1-XBP1 signalling pathway of the UPR was found to be involved in the expression of BDNF in neurons, resulting in promotion of dendritic elongation and branching [Saito et al., 2018]. Moreover, exogenous treatment of BDNF also drives the expression of neurotrophin via IRE1-XBP1 signalling cascades. For these reasons we concluded that BDNF release could be independent on the glutamate system, but we agree that this issue might be more complex therefore we changed the sentence as follows “S-ketamine played the key role in increasing the release of brain-derived neurotrophic factor (BDNF), indicating that the drug has a complex mechanism of action in astrocytes.”

Zhou, B.; Zuo, Y.X.; Jiang, R.T. Astrocyte morphology: Diversity, plasticity, and role in neurological diseases. CNS Neurosci Ther. 2019, 25, 665-673.

Yu, W.; Fang, H., Zhang, L.; Hu, M.; He, S.; Li, H.; Zhu, H. Reversible Changes in BDNF Expression in MK-801-Induced Hippocampal Astrocytes Through NMDAR/PI3K/ERK Signaling. Front Cell Neurosci. 2021,15, 672136.

Manahan-Vaughan, D.; von Haebler, D.; Winter, C.; Juckel, G., Heinemann, U. A single application of MK801 causes symptoms of acute psychosis, deficits in spatial memory, and impairment of synaptic plasticity in rats. Hippocampus 2008, 18, 125–134.

Saito, A.; Cai, L.; Matsuhisa, K.; Ohtake, Y.; Kaneko, M.; Kanemoto, S.; Asada, R.; Imaizumi K. Neuronal activity-dependent local activation of dendritic unfolded protein response promotes expression of brain-derived neurotrophic factor in cell soma. J Neurochem. 2018, 144, 35-49.

Introduction:

Lines 53-60 were embedded in the ms text but should be part of Figure 1 legend

Lines 53-60 have been corrected

Lines 63-64 “It has also been shown that ER stress is also associated with neurodegenerative diseases

Lines 63-64 have been corrected

Lines 76-87: theories of depression and drug treatment are not properly presented. The most widely recognized theory is based on neuroplasticity and must be adequately discussed as an evolution of the monoaminergic hypothesis. Confusion must be avoided. Moreover, ketamine can not be put on the same level as TCA!! Racemic ketamine has been introduced as off-label treatment for TRD!

According to the Reviewer’s suggestion we changed the paragraph about theories of depression and drug treatment, to avoid confusion. We would like to mention that we didn’t put on the same level ketamine and TCAs. We only wanted to point the drugs which are currently used in the treatment of refractory depression but we agree that it was written not clearly enough. We revised the paragraph and hope it is much more understandable now.

Lines 90-91: The study from Bown et al. seems to highlight a role for alterations in the UPR pathway in association with suicide more than with depression. I would instead give more space to previous preclinical evidence.

Many animal models of depression and human studies show elevated brain ER stress response (UPR pathway). The link between ER stress and depression has been discussed in more details according to the Reviewer’s suggestion.

Results:

Reporting all the qPCR results in tables does not help in their understanding. I would suggest to consider to graph at least the significant ones.

We showed the most significant qPCR results in a graph form according to the Reviewer’s and Editor’s suggestions. The previous table containing all results is attached in supplementary materials.

Round 2

Reviewer 2 Report

I thank the authors for their effort in improving the quality of the ms.